# Dynamic Hormone Gradients Regulate Wound-Induced de novo Organ Formation in Tomato Hypocotyl Explants

**DOI:** 10.3390/ijms222111843

**Published:** 2021-10-31

**Authors:** Eduardo Larriba, Ana Belén Sánchez-García, María Salud Justamante, Cristina Martínez-Andújar, Alfonso Albacete, José Manuel Pérez-Pérez

**Affiliations:** 1Instituto de Bioingeniería, Universidad Miguel Hernández, 03202 Elche, Spain; elarriba@umh.es (E.L.); ana.sanchezg@umh.es (A.B.S.-G.); mjustamante@umh.es (M.S.J.); 2CEBAS-CSIC, Department of Plant Nutrition, Campus Universitario de Espinardo, 30100 Murcia, Spain; cmandujar@cebas.csic.es (C.M.-A.); alfonsoa.albacete@carm.es (A.A.)

**Keywords:** de novo shoot apical meristem regeneration, de novo root regeneration, time course RNA-Seq, hormone balance, U-HPLC-HRMS

## Abstract

Plants have a remarkable regenerative capacity, which allows them to survive tissue damage after biotic and abiotic stresses. In this study, we use *Solanum lycopersicum* ‘Micro-Tom’ explants as a model to investigate wound-induced de novo organ formation, as these explants can regenerate the missing structures without the exogenous application of plant hormones. Here, we performed simultaneous targeted profiling of 22 phytohormone-related metabolites during de novo organ formation and found that endogenous hormone levels dynamically changed after root and shoot excision, according to region-specific patterns. Our results indicate that a defined temporal window of high auxin-to-cytokinin accumulation in the basal region of the explants was required for adventitious root formation and that was dependent on a concerted regulation of polar auxin transport through the hypocotyl, of local induction of auxin biosynthesis, and of local inhibition of auxin degradation. In the apical region, though, a minimum of auxin-to-cytokinin ratio is established shortly after wounding both by decreasing active auxin levels and by draining auxin via its basipetal transport and internalization. Cross-validation with transcriptomic data highlighted the main hormonal gradients involved in wound-induced de novo organ formation in tomato hypocotyl explants.

## 1. Introduction

A recurrent theme in developmental biology is the study of regeneration of lost structures or de novo organ formation after wounding. One of the fathers of Genetics, Thomas Hunt Morgan, spent a few years of his early career studying regeneration on a diversity of organisms [1]. Most of our current knowledge about organ regeneration in plants derives from the study of the *Arabidopsis thaliana* model [2]. During indirect tissue-culture regeneration, the sequential incubation of tissue explants on different hormone-supplemented media reprograms some cells to form new organs (i.e., roots, shoots, or embryos) via specific downstream transcriptional networks [3,4]. Excision-induced adventitious root (AR) formation in detached leaves has also been intensively studied, and some of the hormonal and transcriptional networks that are involved in this process have been recently identified [5,6]. In both experimental systems though, the establishment of differential hormone-responsive patterns across the regeneration zone of the explants is critical for de novo organ initiation [7].

In a previous study [8], we characterized wound-induced AR formation in young explants of *Solanum lycopersicum* ‘Micro-Tom’ through physiological and genetic approaches. In our working model, a regulated polar auxin transport (PAT) from the shoot and local auxin biosynthesis triggered by the wounding were both required for the restoration of internal auxin gradients. In turn, these endogenous gradients trigger the de-differentiation and cell cycle reactivation of some cambial cells in the vasculature, which preceded the development of ARs. Our results using hypocotyl explants of different lengths favored the hypothesis that an endogenous pre-established auxin response gradient within the hypocotyl is established before wounding. As a result, the highest auxin responses were observed on the most basal region of the hypocotyl which determine subsequent AR formation in this region, and de novo shoot formation was later observed in the apical region of the explants [8,9]. Besides, we performed detailed transcriptome and targeted metabolomics analyses and found metabolic differences and divergent developmental pathways that contributed to the observed tissue-specific organ formation responses of ‘Micro-Tom’ hypocotyl explants [9]. Our results indicate that callus growth in the apical region of the hypocotyl, which precedes de novo shoot formation, depends on a specific metabolic switch involving the upregulation of the photorespiratory pathway and the differential regulation of genes involved in photosynthesis and glycolysis [9].

To investigate the hormonal regulation of wound-induced de novo organ formation in tomato hypocotyl explants, we performed targeted profiling of several phytohormone-related metabolites on different regions of the hypocotyl and at different times after excision and found that endogenous levels of several hormones changed shortly after root and shoot excision. Cross-validation with previous RNA-seq data [9] allowed us to confirm that genes related to auxin and cytokinin (CK) crosstalk were dynamically regulated during de novo organ formation both in the apical and the basal region of the explants within a short developmental window, which contributed to the observed region-specific regenerative responses.

## 2. Results

### 2.1. Shoot Regeneration Is Correlated with AR Formation in Tomato Hypocotyl Explants

We described previously that a pre-pattern of a rootward gradient of both elevated auxin responses and auxin concentration limited wound-induced AR formation in ‘Micro-Tom’ shoot and hypocotyl explants to a narrow region within the basal end of the tissue near the wounding site [8]. To characterize the regenerative capacity of hypocotyl fragments without the shoot (Figure 1a), we quantified the extent of de novo shoot formation and found that higher regeneration levels were observed on the apical region of fragments that already included the most basal region of the hypocotyl, irrespectively of the length of the explants (Figure 1b,c). In addition, we found high and positive correlations between the areas of the AR system and the regenerated shoot in those explants excluding the most basal region of the hypocotyl (Figure 1d), suggesting that de novo organ formation was developmentally and spatially regulated.

### 2.2. Time Course and Spatial Gradient of Hormone Accumulation during Wound-Induced Regeneration

Separation of stem cuttings from the mother plant alters the endogenous hormonal homeostasis in the basal region of the cuttings, with functions in wound response and during root induction [10,11]. We quantified the endogenous hormone levels during wound-induced organ formation in different regions of tomato hypocotyl explants through targeted metabolomics (Figure 2a,b and Appendix A). Indole-3-acetic acid (IAA), the most active auxin, displayed an intriguing distribution along the apical-basal axis of the hypocotyl before wounding (0 hae), with the lowest levels found in the most basal region of the hypocotyl (Figure 2b). Tissue-specific gradients of endogenous IAA were observed upon excision (Figure 2c). Low and constant IAA levels were observed in the apical region of the hypocotyl. The central region of the hypocotyl contained the highest IAA levels at excision time, which were rapidly reduced thereafter. A transient increase of endogenous IAA was observed in the basal region of the hypocotyl after excision and up to 10 hae (Figure 2c). On the other hand, *trans*-zeatin (*t*Z) levels were not significantly changed along the apical-basal axis of the hypocotyl up to 96 hae, when a shootward gradient of *t*Z was observed (Figure 2d). Interestingly, when we cut the hypocotyl explants in half (DC and BA; Figure 2a), *t*Z levels were higher in the apical region of these explants at 24 hae, suggesting the rapid activation of *t*Z biosynthesis in the apical region (Figure 2e). Regarding gibberellins, GA3 displayed a conspicuous shootward gradient at 0 hae, it was more abundant than GA4 and, its levels diminished during the time course (Appendix A).

We measured the ethylene precursor 1-aminocyclopropane-1-carboxylic acid (ACC) and found that ACC steadily increased upon excision following a steep shootward gradient, with the highest ACC amount observed at 24 hae in the apical region of the explants (Appendix A). Abscisic acid (ABA) levels were low in the basal region of the hypocotyl explants during the time course (Appendix A). Jasmonate (JA) displayed a strong shootward gradient that was reduced after wounding and during the time course (Appendix A), whereas methyl-JA (MeJA) dynamically increased at 6 and 10 hae, mainly in the basal region of the explants (Figure 2f). Besides, salicylic acid (SA) levels were low at 0 hae, and steadily increased during the time course in all regions (Appendix A). Low levels of the strigolactone solanacol (SL) and of the brassinosteroid (BR) epibrassinolide (EpiBL) were detected (Appendix A–h), and EpiBL showed significant increased levels at 6 and 10 hae in the basal region of the explants (Appendix A).

Taken together, our results indicate that high levels of IAA, MeJA, and EpiBL were observed shortly (6–10 h) after excision in the basal region of the explants on a steep rootward gradient, whereas increased levels of *t*Z and ACC were induced upon wounding on a shootward gradient.

### 2.3. Mild Deregulation of Auxin-Homeostasis Genes during Wound-Induced Regeneration

Local application of 1-naphthalene acetic acid (NAA) in the apical region of whole hypocotyl explants (NAAd) restored AR formation to that of the shoot explants (Figure 3a,b). Additionally, local application of NAA in the basal region of the hypocotyl explants (NAAm) caused tissue over proliferation (i.e., callus formation) and a mild increase in rooting (Figure 3b,c). A similar phenotype was observed after the incubation of hypocotyl explants with 2-naphthoxyacetic acid (2-NOA) (Figure 3b,c), a known inhibitor of auxin influx transporters [12], and callus growth in the basal region of 2-NOA-treated explants correlated with a slight reduction (*p*-value = 0.05) of shoot regeneration (Figure 3d and Appendix A). These data confirmed our previous results [8] and suggested that auxin-induced callus formation in the basal region of the explants delayed de novo shoot formation.

We gathered data from a previous RNA-seq experiment [9] and found 129 expressed genes related to the auxin pathway, 95 of which were deregulated (Appendix A). Enrichment of upregulated differentially expressed genes (DEGs) related to auxin homeostasis and auxin transport was observed in both regions of the hypocotyl explants during the time course (Appendix A). Genes related to auxin responses were enriched among the downregulated DEGs (Appendix A). Cell-to-cell auxin transport is dependent on the spatial arrangement of AUX/LAX influx carriers [13], PIN-FORMED (PIN) efflux carriers, and ATP-binding cassette B (ABCB) transporters [14]. In addition, the PIN-LIKES (PILS) proteins, which mostly localize to the endoplasmic reticulum, attenuate cellular auxin responses by reducing cytoplasmic IAA [15]. We found a significant downregulation of *AUX/LAX* genes in the apical region, mainly between 24 and 192 hae (Figure 3e). Several genes (Solyc11g067300, Solyc06g009280, and Solyc02g087410) encoding ABCB proteins previously associated with auxin efflux [16] (*ABCB1A*, *ABCB4E*, and *ABCB15D*, respectively), were upregulated during the time course in both regions (Figure 3e). In contrast, Solyc02g087870, encoding the *ABCB19* auxin transporter associated with AR induction in the Arabidopsis hypocotyl after wounding [17], was expressed at higher levels in the basal region after wounding and was downregulated in the apical region (Figure 3e). We confirmed by quantitative reverse transcription PCR (qRT-PCR) that *ABCB15D* levels increased at 24 hae as regards 0 hae; additionally, its expression was highly upregulated at 6 hae, with similar levels in the apical and the basal regions (Figure 3f). Most *PIN* genes were downregulated in the apical region of the explants during the time course, whereas a few of them (Solyc03g118740, Solyc10g078370, and Solyc04g056620) were upregulated in the basal region (Figure 3e). Besides, two tomato *PILS1 and PILS2* genes (Solyc04g082830 and Solyc02g082450) were upregulated in the apical region after wounding and in the basal region at 24 hae (Figure 3e). These results suggest regulation of PAT leading to differential auxin mobilization in the apical and the basal region during de novo organ formation.

### 2.4. Spatial and Temporal Regulation of Auxin Levels during Wound-Induced Organ Formation

IAA is rapidly catabolized through oxidation via peroxidases to indole-3-metanol (IMet) [18] or via DIOXYGENASE FOR AUXIN OXIDATION 1 (DAO1) enzymes to oxindole-3-acetic acid (oxIAA) [19]. In addition, IAA could be reversibly or irreversibly conjugated to several amino acid residues [20]. Furthermore, IAA could be methylated, and the methyl-IAA ester (MeIAA) is thought to be required for the fine-tuning of specific auxin responses [21,22]. To investigate auxin homeostasis regulation [23], we measured several IAA-precursors and metabolites (Appendix A), some of which already displayed endogenous shootward (e.g., indole-3-pyruvic acid; IPyA) or rootward (e.g., Tryptophan; Trp) gradients along the apical-to-basal axis before wounding (0 hae; Figure 4a). Trp levels steadily increased after wounding mostly in the basal region (Appendix A), and the IAA direct precursor IPyA slightly increased at 6 hae in the basal region of the hypocotyl explants and decreased between 0 and 24 hae in the apical region (Figure 4b). In agreement with these results, we found that genes encoding the first steps of Trp biosynthesis from chorismate were upregulated after wounding and that the genes encoding Trp aminotransferases (TAA1/TAR) and YUCCA (YUC) family of flavin monooxygenases (Figure 4c) required for IAA biosynthesis [24] were also deregulated.

We found that IMet levels steadily increased during the studied time course (Appendix A), while oxIAA levels were not significantly changed (Appendix A). MeIAA levels significantly increased in the basal region at 6 hae and 10 hae, mirroring the IAA levels (Figure 4d). We found one expressed gene (Solyc07g064990), encoding a putative ortholog of IAMT1 of *A. thaliana* [22], that was specifically upregulated in the basal region after wounding, while the putative ortholog (Solyc06g048570) encoding the methyl esterase MES17 [25], was downregulated (Figure 4c). We measured the endogenous levels of IAA conjugated to isoleucine (Ile) and aspartate (Asp) residues as we previously found that they negatively correlated with rooting performance in carnation stem cuttings [26,27]. IAA-Asp and IAA-Ile levels displayed similar profiles in the basal region and the central region of the hypocotyl explants, with a sharp decrease between 0 and 24 hae in both regions (Figure 4e and Appendix A). IAA-Ile levels in the apical region of the hypocotyl explants were low and did not significantly change during the time course (Appendix A), while IAA-Asp levels in the apical region highly increased at 6 hae (Figure 4e). Both in whole (DCBA) or half-sectioned (DC or BA) hypocotyl explants, we found significantly higher levels of IAA-Asp and IAA-Ile in the basal region of the explants specifically at 24 hae (Figure 4f and Appendix A). These results suggested tight regulation of IAA levels through its reversible (IAA-Ile) and irreversible (IAA-Asp) conjugation to amino acid residues that is both tissue-specific and temporally dependent. Indeed, we found specific downregulation in the basal region from 24 hae onwards of a gene, Solyc02g064830 (Figure 4c), encoding the GH3.1 enzyme putatively involved in IAA-Asp conjugation [28]. Besides, Solyc12g005310 and Solyc07g063850, encoding the GH3.5 and GH3.6A enzymes involved in fine-tuning the hormone crosstalk during AR formation in Arabidopsis hypocotyls [29], were strongly downregulated in the apical region from 24 hae onwards (Figure 4c and Appendix A). Regarding the apical region, the Solyc02g064830 gene encoding a GH3.1 enzyme was upregulated at 192 hae (Figure 4c). Indeed, we confirmed by qRT-PCR the *GH3.6A* downregulation at 24 hae in the basal regions, and at 6 and 24 hae in the apical region (Figure 4g).

### 2.5. Functional Validation of the Endogenous Auxin Response Gradient during De Novo Organ Formation

We identified 65 expressed genes encoding components of the SCF^TIR1/AFB^ complex, as well as of the Aux/IAA corepressors and the ARF transcription factors involved in auxin transduction [30] (Appendix A). Only 8 genes were found deregulated among those encoding proteins involved in SCF^TIR1/AFB^ function (Figure 5a). Most of the expressed *ARF* genes [31] were only slightly deregulated in our RNA-Seq, with some exceptions (e.g., *ARF5* and *ARF10A*; Figure 5b). In addition, we found 20 DEGs encoding the Aux/IAA corepressors [32] whose expression was constitutively downregulated at higher levels (>2-fold) as regards 0 hae and mainly in the apical region (Figure 5c). Conversely, a subset of Aux/IAA genes (e.g., *IAA2*, *IAA11*, and *IAA15*) were upregulated in the basal region at specific time points after wounding (Figure 5c).

We previously found that the chemical inhibition of YUC enzymes with yucasin DF (YDF) reduced the rooting capacity of half-sectioned hypocotyl explants [8]. We now treated the whole hypocotyl explants with L-kynurenine, which inhibits TAA1/TAR activity [33], and found a slight but significant delay in AR emergence and rooting capacity (Appendix A), confirming that local IAA biosynthesis also contributed to AR formation after wounding in the absence of any auxin transported from the shoot. Whole hypocotyl explants treated with auxinole for 48 h, which blocked the formation of the TIR1–IAA–Aux/IAA complexes [34], showed a slight reduction in rooting capacity as regards the mock-treated ones (Figure 5d). Both the regeneration response in the basal region of the explants and the number of AR primordia visualized by *DR5::GUS* marker expression (Figure 5e) were slight but significantly enhanced after the auxinole treatment (Figure 5f,g). Interestingly, we found that auxinole did not significantly change the distance from the wounding site where the AR primordia are formed (L0) nor the length of the AR formative region (L1) (Figure 5h). The continuous incubation of apical hypocotyl fragments with auxinole significantly enhanced rooting capacity although the ARs were much shorter, while the combined treatment with auxinole and YDF of these explants caused an intermediate phenotype (Appendix A).

Taken together, these results confirm that strict regulation of the temporal window of auxin response in a defined location within the basal region of the explants is required to effectively induced AR formation and that other unknown (i.e., non-auxin) factors also contribute to root founder cell activation after wounding.

### 2.6. CK Levels Are Dynamically Regulated during De Novo Organ Formation

In the hypocotyl explants, *t*Z was the most abundant CK, although we also detected *cis*-zeatin (*c*Z), dihydrozeatin (DHZ), and 2-isopentenyladenine (2-iP) at different levels (Appendix A). Whole CK levels (*t*Z, *c*Z, DHZ and 2-iP) steadily increased in the apical region of the explants after excision and during the time course (Figure 6a). The lowest CK levels were found in the basal region of the explants at 6 and 10 hae (Figure 6a). Intriguingly, *c*Z levels fluctuated during the time course in the studied regions (Figure 6b). Before wounding (0 hae), *c*Z levels were higher in the central and the basal regions of the hypocotyl explants than in the apical region, and they rapidly diminished in the central and basal regions only during a short period (Figure 6b). The highest *c*Z levels after wounding were found both in the apical and the basal region at 24 hae, while its levels reached their minimum in the basal region of the explants at 6 hae (Figure 6b). Similar fluctuating *c*Z levels were also found when we cut the hypocotyl explants in half (Appendix A).

We found 97 expressed genes related to CKs (Appendix A) with a higher enrichment of deregulated genes involved in CK biosynthesis and CK response (Appendix A). Several ISOPENTENYLTRANSFERASE (IPT)-encoding genes, *IPT2*, *IPT3,* and *IPT4* (Solyc04g007240, Solyc05g009410, and Solyc09g064910), were upregulated from 24 hae onwards in both regions (Appendix A). Two CYP735A-encoding genes (Solyc02g085880 and Solyc02g094860) were specifically upregulated at 24 hae in both regions (Appendix A). Additionally, the expression of *LONELY GUY* (*LOG*) genes, involved in the direct activation pathway of CKs [35], were mostly downregulated in both regions and during the time course, as regards their highest levels at 0 hae (Appendix A). Several genes encoding glycosyltransferases putatively involved in the reversible inactivation of CKs [36] were also deregulated in our RNA-seq at different time points after wounding in both regions (Appendix A).

To confirm that endogenous CK accumulation contributes to de novo organ formation, we used lovastatin, a potent inhibitor of the mevalonate pathway to interfere with CK cytosolic biosynthesis (see Materials and Methods) [37]. Incubation of whole hypocotyl explants with 125 µM of lovastatin interfered with AR emergence (Figure 6c). Interestingly, rooting capacity at 17 dae was significantly enhanced on hypocotyl explants grown on 125 µM lovastatin for five days (Figure 6d), while their shoot regeneration was strongly delayed (Figure 6e). To evaluate the effect of CK inhibition on AR initiation, we visualized *DR5::GUS* expression after 48 h on 125 µM lovastatin (Figure 6f). We found that lovastatin treatment enhanced both the AR regeneration response (Figure 6g) and the number of *DR5::GUS* foci (Figure 6h). However, in sharp contrast with that found in the auxinole treatment (see above), lovastatin significantly enhanced the length of the AR formative region (Figure 6i).

Long-distance CK transport from the root to the shoot via the xylem is essential for shoot growth regulation [38]. We found that a gene encoding the ABCG14 transporter, Solyc08g075430, whose Arabidopsis ortholog is essential for root-to-shoot CK translocation [39], was highly expressed in the basal region from 24 hae onwards (Appendix A). CKs are perceived by membrane-localized histidine-kinase (AHK) receptors and are transduced through His-Asp phosphorelay proteins (AHP) to activate B-type response regulators (RR). Type-B RR (ARR-B) translocate to the nucleus and directly activate the expression of the type-A RR (ARR-A) negative regulators, which are further stabilized by AHP phosphorylation [40]. *AHK4* (Solyc07g047770) and *AHP1* (Solyc01g080540) genes were downregulated in both regions after wounding (Appendix A). Regarding the ARR-B encoding genes [41], *ARR-B5* (Solyc05g014260 expression was downregulated from 24 hae onwards, mostly in the apical region (Appendix A). *ARR-A16* (Solyc06g048930) and *ARR-A16/17* (Solyc06g048600) expression was significantly upregulated in the apical region after wounding and downregulated in the basal region (Appendix A). Additionally, *ARR-A5/7/15* (Solyc03g113720) expression was significantly reduced in the basal region of the hypocotyl explants from 24 hae onwards (Appendix A). We studied the expression levels of Solyc03g113720 and Solyc06g048930 using qRT-PCR (see Materials and Methods) and found the highest upregulation of both genes in the apical region of the explants at 6 hae (Appendix A). These results suggest differential regulation of CK responses in basal and apical regions during de novo organ formation.

### 2.7. A Functional Auxin-to-CK Endogenous Gradient Controls De Novo Root Regeneration after Wounding

Callus formation relies on the endogenous auxin-to-CK ratio [42]. Based on the hormone profiling results observed during the studied time course (Appendix A), the highest auxin-to-CK ratio was found in the basal region of the explants at 6 and 10 hae (Appendix A). Conversely, the apical region of the hypocotyl explants contained low auxin-to-CK ratios during the studied time course, but the lowest auxin-to-CK ratio was found in the basal region of the hypocotyl before wounding (Appendix A). Our DEG enrichment analysis showed that genes related to CKs were more deregulated than those related to auxins upon excision (Appendix A). Downregulated DEGs related to auxin were three-fold enriched in the apical region of the explants compared to those in the basal region (Appendix A). Besides, upregulated DEGs related to CK were highly enriched in the apical region of the explants after wounding. These results validate the low auxin-to-CK ratio found in the apical region upon excision.

To investigate the relevance of the observed auxin-to-CK ratio for wound-induced de novo organ formation, we studied the effect of exogenously altering this ratio (see Materials and Methods). Incubation of whole hypocotyl explants with 0.1 µM *t*Z caused a significant delay in AR emergence (Figure 7a), and a reduction in the rooting capacity of the explants (Figure 7b). Moreover, the NAA treatment induced proliferation of the basal region of the hypocotyl and enhanced rooting capacity (Figure 7b,c). In addition, *de novo* shoot formation on *t*Z treated explants was held at stage 1 (Figure 7d). Combining NAA and *t*Z resulted in an intermediate phenotype as regards AR emergence and rooting capacity (Figure 7a–c), and in mild enhancement of de novo shoot formation (Figure 7d). A combined auxinole and lovastatin treatment for 48 h did not significantly enhance the number of *DR5::GUS* foci in the basal region of the hypocotyl (Figure 7e,f), nor the length of the AR formative region (Figure 7g). These results suggest that CKs have a dual role, one as a positive regulator of cell proliferation in the AR formative region near the wound (likely independent of the auxin signal itself), and another as a negative regulator of auxin responses in the shootward region of the hypocotyl near the wound and above the AR formative region.

## 3. Discussion

To enhance our understanding of de novo organ formation in plants, we established a new experimental system using ‘Micro-Tom’ hypocotyl explants after shoot and root excision, as these explants can regenerate the missing structures without the exogenous application of plant hormones [8,9]. We found that endogenous hormone levels were layered along the apical-basal axis of the hypocotyl before wounding (this work). A clear shootward gradient, with maximum levels in the apical region of the hypocotyl below the cotyledons, was found for GA3, JA, and ABA. Furthermore, inactive IAA conjugates, such as IAA-Asp and IAA-Ile, accumulated on a sharp rootward gradient. Interestingly, we found a peculiar rootward-like accumulation pattern for IAA, MeIAA, and MeJA before wounding, as the minimum levels of these compounds were found in the most basal region of the hypocotyl. Other hormones, such as *t*Z or SA, displayed similar levels along the apical-basal axis of the hypocotyl before wounding. The observed endogenous gradients (e.g., GA3, IAA, and MeJA) might provide endogenous cues for setting up the positional information along the apical-basal axis of the hypocotyl, which in turn determines differential developmental responses along this axis, such as hypocotyl growth in its apical region [43] or the development of hypocotyl-derived and anchor roots [44] in its most basal region. Endogenous hormone levels dramatically changed shortly after root and shoot excision following region-specific patterns. In the most basal region of the hypocotyl explants between 6 and 10 hae, there was a clear overlap of a transient increase in IAA, MeIAA, MeJA, and EpiBL with the concomitant reduction in IAA-Asp, IAA-Ile, and *c*Z.

Several pieces of evidence indicate an IAA and MeJA crosstalk during root specification and growth in Arabidopsis. On the one hand, exogenously-added MeJA regulated auxin redistribution through enhanced IAA biosynthesis [45] and PAT [46]. On the other hand, ARF6- and ARF8-mediated auxin signaling inhibited endogenous MeJA accumulation by inducing its GH3-mediated inactivation during light-induced AR formation in hypocotyls [29,47]. Although the ‘Micro-Tom’ tomato cultivar carries a mutation in the *DWARF* gene required for BR biosynthesis, this mutation is likely hypomorphic and causes a weak BR-deficient phenotype [48], which is consistent with the low levels of endogenous EpiBL that were observed during de novo organ formation in the hypocotyl (this work). However, the transient increase of EpiBL in the most basal region of the hypocotyl between 6 and 10 hae might contribute to AR initiation in this region, likely through its crosstalk with IAA. Indeed, it has been shown recently that ARF6 and ARF8 also promote the expression of *DWARF4* in the leaf epidermis and that subsequent BR signaling increases cell wall plasticity resulting in directional cell growth [49]. Hence, the observed MeJA and EpiBL gradients might be the direct read-out of the IAA gradient in the most basal region of the hypocotyl after wounding. Alternatively, MeJA and EpiBL might be upstream of IAA levels. Evidence from the latter has been recently reported and revealed a dual effect of BR on auxin, with BR simultaneously promoting auxin biosynthesis and repressing auxin transcriptional output, which has direct effects on root meristem maintenance and auxin-dependent root meristem regeneration [50]. The characterization of the endogenous hormone profile of JA-insensitive and BR-defective mutants in the ‘Micro-Tom’ background [51] will allow us to determine the functional relevance of the dynamic MeJA and EpiBL gradients observed during wound-induced de novo organ formation.

In tomato shoot explants, the auxin produced in the shoot and the cotyledons is transported towards the basal region of the hypocotyl near the wound, where is required for AR formation [8]. Hypocotyl explants lacked these two endogenous IAA sources, and hence displayed a delay in AR emergence and lower rooting capacity than the shoot explants [8]. Despite the internal structure and the hormonal profile within the hypocotyl differ significantly along their apical-basal axis, both apical and basal explants regenerate new root and shoot organs (this work). De novo shoot formation always occurred, and it was dependent on the development of the AR system, suggesting that some root-derived compounds are required for shoot apical meristem (SAM) initiation. One such candidates are CKs, which are known to promote tissue proliferation and to regulate *WUSCHEL* (*WUS*) expression in the SAM both in planta and during indirect tissue-culture regeneration [52,53]. *t*Z levels significantly increased in the upper region of the hypocotyl explants at 96 hae, which was consistent with enhanced *WUS* expression in the apical region during the time course [9]. Intriguingly, the expression of genes encoding IPT enzymes involved in the rate-limiting step of CK biosynthesis in Arabidopsis [54], was highly induced in the apical region of the hypocotyl after wounding, suggesting that local CK biosynthesis might also contribute to transient *c*Z accumulation in the apical region after wounding. Additionally, *c*Z accumulation might also contribute to the observed increase in photorespiration in the apical region of the hypocotyl after wounding [9], as it has been shown in *A. thaliana* seedlings that the overexpression of an *IPT* gene induced photorespiration, with a beneficial role during drought stress [55]. In addition to *c*Z, we found high levels of ACC in the apical region of the explants at 24 hae, coinciding with extensive cell proliferation in this region. Downstream candidates for wound-induced cell proliferation are the AP2/ERF transcription factor WOUND INDUCED DEDIFFERENTIATION 1 (WIND1), which enhances endogenous CK responses in Arabidopsis [56], and ANAC071, which is induced by ethylene and that triggered cell proliferation of the cambial initials during wound healing in Arabidopsis flowering stems [57,58]. Additional experiments will be required to address whether similar regulatory networks are involved in wound-induced de novo organ formation in ‘Micro-Tom’ hypocotyl explants and to confirm the pivotal role of photorespiration for adventitious shoot formation [9].

PAT plays a crucial role in excision-induced AR formation in stem cuttings of many species [11], including tomato [8]. In *A. thaliana*, the localized induction of ABCB19 in the hypocotyl after whole root excision leads to enhanced IAA transport and local IAA accumulation near the wound, which in turn triggers AR formation [17]. We found that many genes encoding key PAT components are deregulated in ‘Micro-Tom’ hypocotyl explants in a tissue-specific manner. Most *AUX/LAX* and *PIN* genes were downregulated in the apical region after wounding, while some *PILS* genes were upregulated in this region. Conversely, the expression of most ABCB- and PIN-encoding genes were highly upregulated in the basal region of the hypocotyl shortly after whole root excision, indicating endogenous auxin canalization towards the AR formative region, which was required for AR initiation and further growth. Intriguingly, the central region of the hypocotyl contained high levels of IAA before wounding, while IAA levels were highly reduced in the most basal region of the hypocotyl, likely due to GH3-regulated conjugation of IAA to amino acids, such IAA-Asp and IAA-Ile (this work). In some conditions (high relative humidity, flooding, etc.), tomato could develop ARs in the hypocotyl [59], and we hypothesize that the endogenous reservoir of IAA observed in the central region of the hypocotyl might contribute to this environmentally-regulated AR response. Indeed, the *aerial roots* (*aer*) tomato mutant contains higher IAA levels in the base of the stems due to enhanced PAT, which causes a profuse and precocious formation of AR primordia along the stem [60]. In our experimental system (i.e., ‘Micro-Tom’ hypocotyl explants), whole root excision triggered PAT from the central to the most basal region of the hypocotyl, likely through the enhanced expression of *ABCB19* and *ABCB15D* genes in the basal region of the hypocotyl. *PIN2* upregulation in the basal region of the hypocotyl might contribute to canalize auxin towards the outer cambial AR initials. Moreover, transient GH3 inactivation reduced IAA-Asp and IAA-Ile levels in the basal region of the hypocotyl after wounding and up to 10 hae. Endogenous regulation of auxin levels by GH3 enzymes has been shown to contribute to cultivar-specific wound-induced AR formation in carnation stem cuttings [26], while *GH3.5* overexpression in apple stem cuttings inhibited AR formation by reducing the free IAA content [61]. As a result, a sharp IAA gradient in the vasculature of the basal region of the hypocotyl explants near the wound is produced shortly after wounding, which is in turn required for activation of the AR founder cells in the vascular cambium.

It has been proposed that auxin might act as a morphogen in *A. thaliana* to provide highly reproducible spatiotemporal positional information during organ initiation in the SAM [62] and during root development [63]. Root meristem regeneration after root tip removal in *A. thaliana* is driven by rapid accumulation of auxin near the injury site through local auxin biosynthesis, while PAT is not a crucial player for regeneration in this system [64]. We found a tight regulation of the endogenous levels of IAA-related metabolites between 0 and 24 hae along the apical-basal axis in ‘Micro-Tom’ hypocotyl explants. Based on the observed levels of the direct IAA precursor, IPyA, and on the effect on AR response induced by the auxin biosynthesis inhibitors L-kynurenine and yucasin DF, we hypothesize that local auxin biosynthesis is necessary but not sufficient for wound-induced de novo root formation in tomato hypocotyl explants. Unexpectedly, when TIR1-mediated auxin responses were inhibited by auxinole incubation, proliferation of the inner tissues of the hypocotyl was induced, which resulted on a mild increase in the number of *DR5::GUS* expressing foci. These results suggest that other unknown (i.e., non-auxin) factors contribute to root founder cell activation after wounding. Alternatively, an auxin-regulated negative regulatory loop might restrict root founder cell specification in the vascular cambium near the wound site. A candidate circuit for such regulatory loop might involve ARF7 and IAA18/POTENT, which has been shown to regulate the oscillatory behavior of the pre-branching sites during lateral root formation in Arabidopsis [65]. 

Classical in vitro culture experiments showed how plant tissue differentiation is affected by the exogenous ratio of auxin and CK levels in the growing media [66]. It is well known the inhibitory effect of a low auxin-to-CK ratio during wound-induced AR formation in stem cuttings of several species [11]. Recent studies in *A. thaliana* roots uncovered the molecular mechanisms of CK-mediated auxin gradient refinement [67]. In the primary root, the boundary between proliferating and differentiating cells is controlled by the antagonistic interaction between CKs and the auxin gradient, with CKs contributing to the shaping of this gradient by positioning the auxin minimum through both PAT regulation [68] and local IAA degradation [69]. CKs act through ARR1 in the root meristem, by repressing (via SHY2/IAA3) the expression of several *PIN* genes (*PIN1*, *PIN3,* and *PIN7*), and by the direct activation of two GH3-encoding genes (*GH3.3* and *GH3.17*). Additionally, CK levels in the lateral root cap indirectly control auxin levels in the entire root meristem by the GH3-mediated conjugation of auxin to different amino acids [70,71]. We found that continuous inhibition of CK biosynthesis in ‘Micro-Tom’ hypocotyl explants hindered AR development, whereas a transient inhibition of CK biosynthesis enlarged the AR formative region towards the shootward direction resulting in a higher number of ARs. Our results suggest that local CK biosynthesis in the basal region of the ‘Micro-Tom’ hypocotyl explants plays an inhibitory role in AR induction by restricting auxin-induced formative responses to a defined region near the wounding. Hence, we hypothesize that the observed expansion of hypocotyl-derived ARs in the shoot explants of some tomato cultivars, such as ‘Moneymaker’ and ‘Heinz 1706-BG’ [8], could be caused by the local alteration of the endogenous auxin-to-CK ratio, which can now be tested.

## 4. Materials and Methods

### 4.1. Plant Material and Growth Conditions

Seedlings of the tomato cultivar ‘Micro-Tom’ were grown in vitro as described elsewhere [8]. Hypocotyl explants were obtained after removing the whole root system (2–3 mm above the hypocotyl-root junction) and the shoot apex (just below the cotyledons’ petiole) with a sharp scalpel, at the 100–101 growth stages [72], hereafter 0 days after excision (dae). Hypocotyl explants were transferred to 120 × 120 mm (length × width) sterile Petri dishes containing 75 mL of standard growing medium (SGM) as previously described [8].

### 4.2. Macroscopic Studies of Wound-Induced AR Formation and Shoot Regeneration

Hypocotyl explants were incubated for 17 days in 120 × 120 mm Petri dishes containing 75 mL of SGM plates supplemented with 0.3 µM NAA (Duchefa Biochemie, Haarlem, The Netherlands), 40 µM 2-NOA (Merck, Burlington, MA, USA), 0.1 µM *t*Z (Duchefa), 20–50 µM auxinole or 50 µM YDF. For analysis of CK inhibition, hypocotyl explants were transferred to sterile multi-well plates containing 1 mL of liquid growing medium supplemented with 125 µM lovastatin (Merck) or 50 µM auxinole for 24 h or 48 h. Then, samples were transferred to SGM or GUS staining solution. For GUS staining, whole hypocotyl explants were processed as described in [8]. Imaging of GUS-stained hypocotyls was obtained on a Motic BA210 microscope (Motic Europe, Barcelona, Spain) with a built-in image capture system Moticam 580INT (Motic Europe, Barcelona, Spain).

### 4.3. RNA Isolation, First-Strand cDNA Synthesis, and qRT-PCR

Total RNA was extracted from three-to-four mm apical and basal regions of eight hypocotyl explants, which were collected at 0, 24, 96, and 192 hae. RNA extractions were performed in triplicate with Direct-zol RNA MiniPrep kit (Zymo Research, Irvine, CA, USA) and DNase treated on column following the manufacturer’s protocol. First-strand cDNA was synthesized with 1 μg of purified RNA using iScript cDNA Synthesis kit (Bio-Rad, Hercules, CA, USA). For qRT-PCR, primers amplified 112–160 base pairs of the cDNA sequences (Appendix A). When possible, forward and/or reverse primers were designed to bound to different exons and to hybridize across consecutive exons to avoid amplifying genomic DNA. Amplifications were prepared as described in [8]. Gene expression analyses were carried out using the 2^−ΔΔCt^ method [73]. The housekeeping *SlACTIN2* (Solyc03g078400) gene was chosen as a reference. For each gene, the mean fold-change values relative to the basal part of ‘Micro-Tom’ hypocotyls at 0 hae were used for graphical representation.

### 4.4. RNA-Seq Analysis

RNA-seq data was obtained from the NCBI SRA database (SRR14598206-SRR14598225) and processed as described in [9]. Briefly, read count normalization and differential gene expression analysis were carried out using DESeq2 integrated into Differential Expression and Pathway analysis (iDEP 9.1) web application [74]. DEGs were filtered using FDR < 0.01 and Log_2_fold change > < |1|. ITAG4.0 gene annotation was retrieved from Sol Genomics Network [75] and Uniprot [76], with manual curation. Their putative *Arabidopsis thaliana* orthologs were identified using BioMart tool from the Emsembl plant database [77], and as well as the identification of orthologous genes between the gene models of in *Arabidopsis thaliana* (TAIR11) and tomato genes models (ITAG4) using Proteinortho software [78]. Additionally, the Kyoto Encyclopedia of Genes and Genomes (KEGG) and PANTHER databases have been used to identify and classify the different genes. Specific sources of gene annotation are indicated in the supplemental tables. DEG enrichment in each category was calculated using the following formula: (number of up or down DEGs in the category/number of all annotated genes in the category)/(number of up or down DEGs in the experiment/number of all expressed genes in the experiment). Heatmap representation was performed using Morpheus [79].

### 4.5. Hormone Extraction and Analysis

Hormone extraction of two independent assays was performed. In the first experiment, whole hypocotyl explants were transferred to SGM keeping their apical-basal polarity. Three replicates including eight hypocotyl explants were collected at 0, 6, 10, 24, 96, and 192 hae; before collection, each hypocotyl explant was sectioned into three fragments (D, BC, and A fragments). In the second experiment, hypocotyls were half-sectioned (DC and BA fragments), and each fragment was transferred to SGM keeping its apical-basal polarity. Three replicates including eight DC or BA fragments were collected at 0, 24, and 72 hae; before collection, each fragment was divided in half and hormone measurements were analyzed in D, C, B, and A fragments. Phytohormones were extracted from frozen tissues and were analyzed as described previously [80,81,82]. Auxin metabolites were identified according to their exact molecular masses and retention times obtained from the total ion chromatograms generated by U-HPLC-HRMS (ThermoFisher Scientific, Waltham, MA, USA) analysis, and they were quantified using analog compounds as standards.

### 4.6. Statistical Analyses

The descriptive statistics were calculated by using the StatGraphics Centurion XV software (StatPoint Technologies, Inc. Warrenton, VA, USA) and SPSS 21.0.0 (SPSS Inc., Chicago, IL, USA) programs. Outliers were identified and excluded from posterior analyses as described elsewhere [8]. We performed multiple testing analyses using the ANOVA F-test or Fisher’s least significant difference methods (*p*-value < 0.01, unless otherwise indicated). Non-parametric tests were used when necessary (i.e., AR emergence and AR number).

## 5. Conclusions

Our results indicate that a small temporal window of high auxin-to-CK accumulation in the basal region of the explants was produced shortly after wounding through a concerted regulation of PAT through the hypocotyl, of local induction of IAA biosynthesis, and of local inhibition of IAA degradation. CKs reached minimum levels in the basal region of the hypocotyl after wounding and up to 10 hae, mostly due to *c*Z reduction. Later, the observed *c*Z increase in this region might contribute to limit the IAA gradient, likely through ARR1-mediated IAA inactivation by GH3 enzymes. In the apical region, though, a minimum of auxin-to-CK ratio is established shortly after wounding (6 hae), both by decreasing active auxin levels via IAA-Asp conjugation and by draining auxin from this region via PAT and auxin internalization by the PILS. As a result, a transient increase in CK response in the apical region of the hypocotyl explants occurs shortly after wounding, as indicated by the increased expression of genes encoding A-type regulators, such as *ARR-5/7/15*, and *ARR-16*. In turn, *ARR-5/7/15* and *ARR-16*, like that found in *A. thaliana* [83], might constrain shoot regeneration to later points (>24 hae), when CK levels rise again. The use of fluorescent sensors for auxin and CK responses, such as *DR5rev::3XVENUS-N7* and *TCSv2::3XVENUS* [84], will allow a detailed study of the auxin and CK crosstalk during wound-induced de novo organ formation and the subsequent identification of the gene regulatory networks involved through fluorescence-activated cell sorting (FACS) coupled to single-cell RNA sequencing (scRNA-seq) [85].

## Figures and Tables

**Figure 1 ijms-22-11843-f001:**
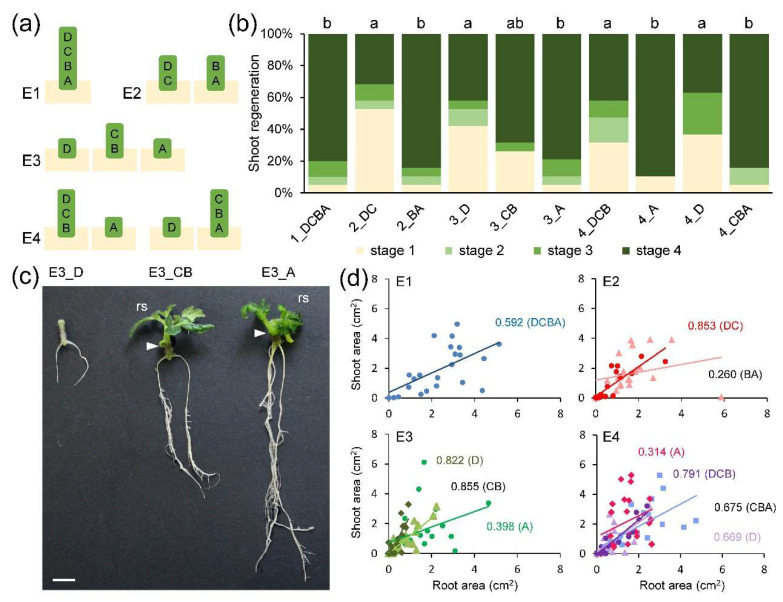
An endogenous developmental gradient regulates de novo shoot formation. (**a**) Diagram showing the hypocotyl fragments used in the different experiments (E1 to E4). (**b**) Shoot regeneration stages of ‘Micro-Tom’ hypocotyl explants at 17 days after excision (dae). (**c**) Representative images of rooted hypocotyl explants at 21 dae. Arrowheads indicate the apical end of the hypocotyl explants at 0 hae; rs: regenerated shoot. (**d**) Relationship between root area and shoot area at 21 dae. Numbers in d indicate Person’s correlation coefficient in the studied samples. Different letters in b indicate significant differences (*p*-value < 0.01) between samples. Scale bars in c: 5 mm.

**Figure 2 ijms-22-11843-f002:**
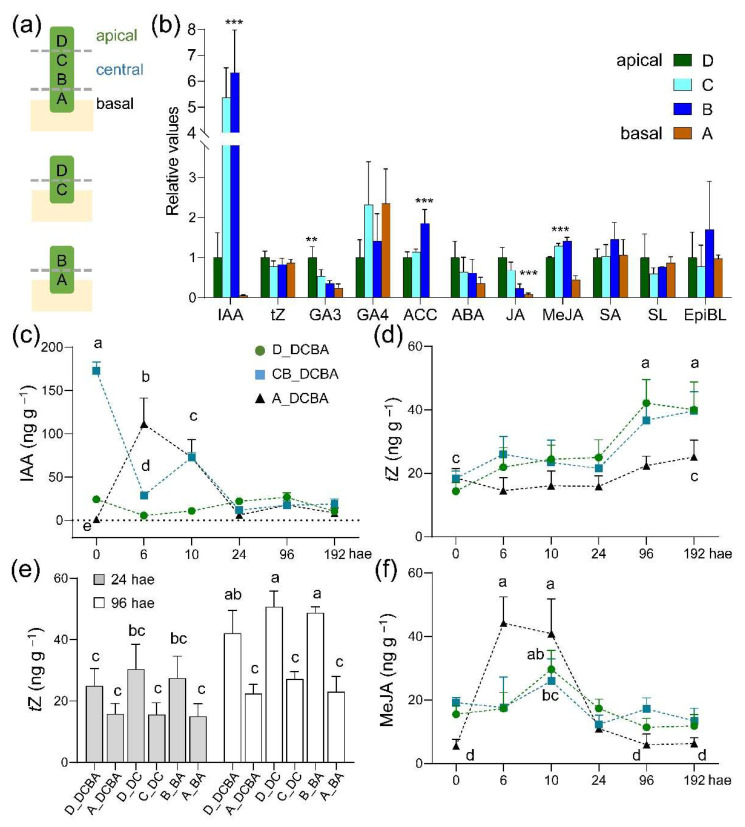
Endogenous hormone levels during de novo organ formation. (**a**) Diagrams showing the hypocotyl fragments used in the different experiments, with dashed lines indicating the regions assayed for hormone levels. (**b**) Relative endogenous levels of several hormones at 0 hae as regards to those in the apical region of the hypocotyl explants; asterisks indicate significant differences (*p*-value < 0.05 *, 0.01 **, or 0.001 ***) between the studied regions (D, C, B, and A) in whole hypocotyl explants (DCBA fragments). (**c**–**f**) Endogenous levels of indole-3-acetic acid (IAA; c), *trans*-zeatin (*t*Z; d-e), and methyl jasmonate (MeJA; f) during the studied time course. (**e**) Endogenous levels of *t*Z at 24 and 96 hae in apical (D) and basal (A) regions of whole (DCBA) or half-sectioned (DC and BA) hypocotyl explants. Different letters in c-f indicate significant differences (*p*-value < 0.01) between the studied samples.

**Figure 3 ijms-22-11843-f003:**
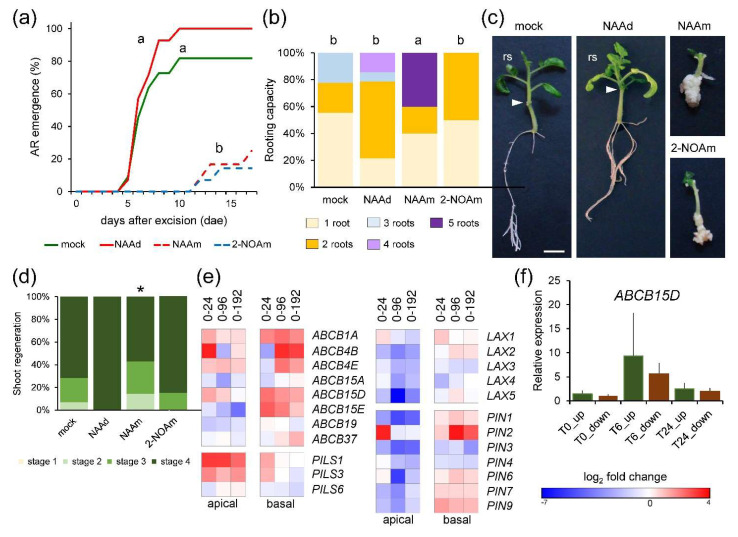
Regulation of auxin pathway genes during de novo organ formation. (**a**,**b**) AR emergence (**a**) and rooting capacity at 14 dae (**b**) of whole hypocotyl explants in response to the indicated treatment; NAAm: 0.3 µM NAA added to the growing medium; NAAd: 0.3 µM NAA added to the distal end of the hypocotyl explants; 2-NOAm: 40 µM 2-NOA added to the growing medium. (**c**) Representative images of the hypocotyl explants at 21 dae. Arrowheads indicate the apical end of the hypocotyl explants at 0 hae; rs: regenerated shoot. (**d**) Shoot regeneration stages of ‘Micro-Tom’ explants at 17 dae. Different letters (**a**,**b**) and asterisk (**d**) indicate significant differences (*p*-value < 0.05) between the treatments. (**e**) Differential expressed genes (DEGs) involved in auxin transport. Expression values of DEGs (false discovery rate [FDR] < 0.01) were in log_2_ fold change relative to 0 hae according to the scale bar in the figure. (**f**) Relative expression levels of *ABCB15D* at 0, 6 and 24 hae in apical (green bars) and basal (brown bars) regions of the whole hypocotyl explants. Scale bars in c: 5 mm.

**Figure 4 ijms-22-11843-f004:**
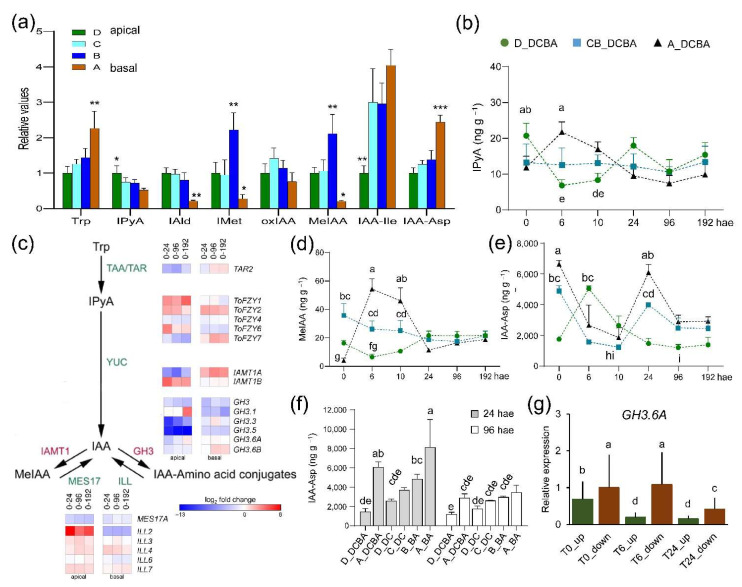
Accumulation of auxin metabolites during de novo organ formation. (**a**) Relative endogenous levels of several auxin precursors and derivatives at 0 hae as regards to those in the apical region of the hypocotyl explants; asterisks indicate significant differences (*p*-value < 0.05 *, 0.01 **, or 0.001 ***) between the studied regions in whole hypocotyl explants (DCBA fragments). (**b**) Endogenous levels of indole-3-pyruvic acid (IPyA) during the studied time course. (**c**) DEGs involved in auxin biosynthesis and homeostasis. Expression values of DEGs (FDR < 0.01) were in log_2_ fold change relative to 0 hae according to the scale bar in the figure. (**d**,**e**) Endogenous levels of methyl-IAA (MeIAA; d), and indole-3-acetyl-L-aspartate (IAA-Asp; e) during the studied time course. (**f**) Endogenous levels of IAA-Asp at 24 and 96 hae in apical (D) and basal (A) regions of whole (DCBA) or half-sectioned (DC and BA) hypocotyl explants. (**g**) Relative expression levels of *GH3.6A* at 0, 6 and 24 hae in apical (green bars) and basal (brown bars) regions of the whole hypocotyl explants. Different letters in b, d-g indicate significant differences (*p*-value < 0.01) between samples.

**Figure 5 ijms-22-11843-f005:**
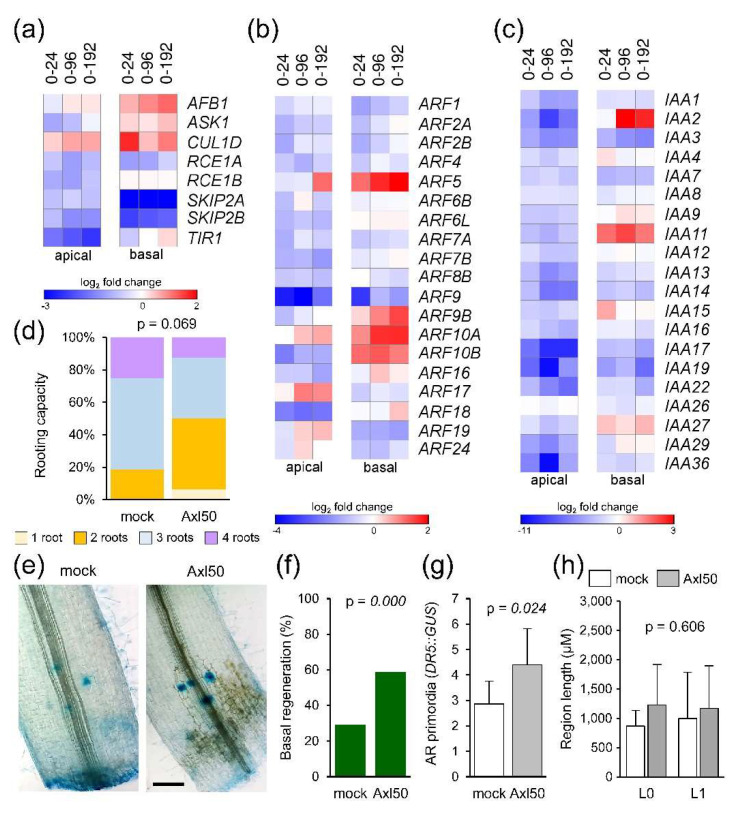
Auxin regulation during de novo organ formation. (**a**–**c**) DEGs involved in auxin signal transduction: SCF^TIR1/AFB^ function (**a**), *ARF* family (**b**), and the *Aux/IAA* family (**c**). Expression values of DEGs (FDR < 0.01) were in log_2_ fold change relative to 0 hae according to the scale bars in the figure. (**d**) Rooting capacity of whole (DCBA) hypocotyl explants at 14 dae; explants were incubated with 50 µM auxinole (Axl50) for 48 h. (**e**) Representative images of *DR5::GUS* expression in the hypocotyl explants after 48 h of mock or 50 µM auxinole (Axl50) treatment. (**f**,**g**) Regeneration response (**f**), and AR primordia expressing *DR5::GUS* (**g**) at 48 h after mock or 50 µM auxinole (Axl50) treatment. (**h**) Length of the distance to the wounding (L0) and of the AR formative region (L1) at 48 h after mock or 50 µM auxinole (Axl50) treatment. Scale bars in e: 500 µm.

**Figure 6 ijms-22-11843-f006:**
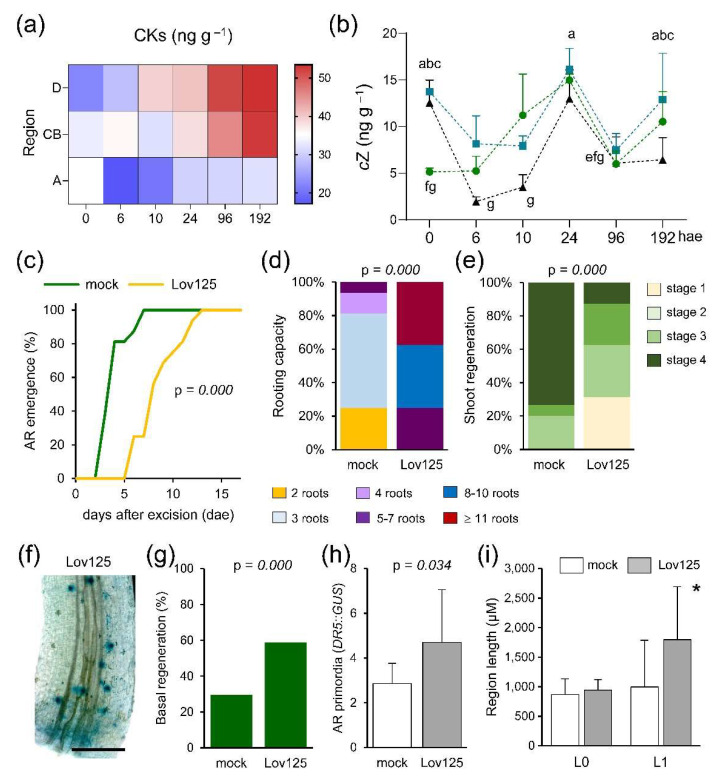
Regulation of CK pathway during de novo organ formation. (**a**) CK profile during the studied time course. (**b**) Endogenous levels of *cis-*zeatin (*c*Z) during the studied time course. (**c**) AR emergence of the studied explants in response to the indicated treatment; Lov125: 125 µM lovastatin added to the growing medium. (**d**) Rooting capacity of ‘Micro-Tom’ explants at 17 dae. (**e**) Shoot regeneration stages of ‘Micro-Tom’ explants at 17 dae. (**f**) Representative image of *DR5::GUS* expression in the hypocotyl explants at 48 h after 125 µM lovastatin (Lov125) treatment. (**g**,**h**) Regeneration response (**g**), and AR primordia expressing *DR5::GUS* (**h**) at 48 h after mock or Lov125 treatment. (**i**) Length of the distance to the wounding (L0) and of the AR formative region (L1) at 48 h after mock or Lov125 treatment. Different letters (**b**) and asterisks (**i**) indicate significant differences (*p*-value < 0.05) between the studied regions in whole hypocotyl explants. Scale bars in f: 1 mm.

**Figure 7 ijms-22-11843-f007:**
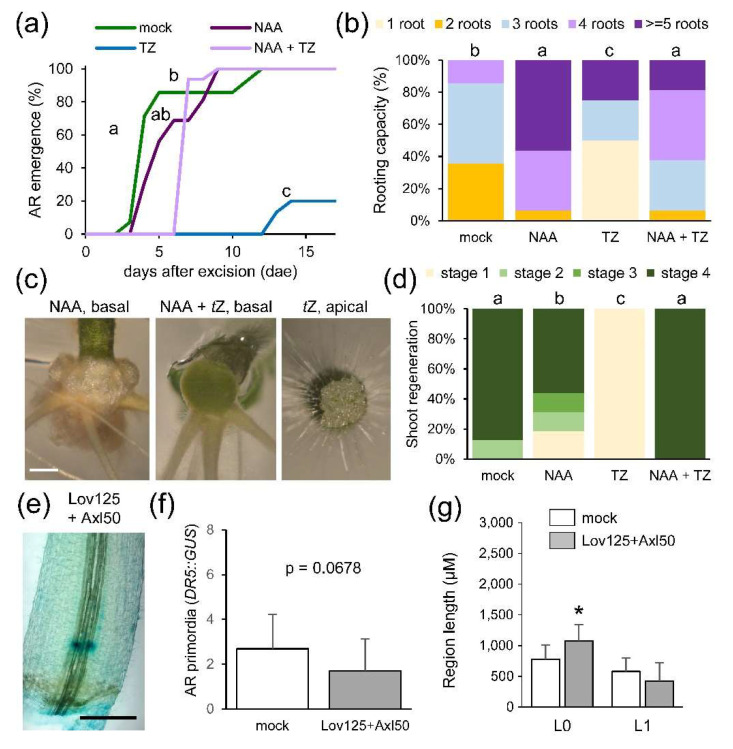
Auxin-to-CK ratio regulates de novo organ formation. (**a**) AR emergence of the studied explants in response to the indicated treatment; NAA: 0.3 µM NAA added to the growing medium; TZ: 0.1 µM *t*Z added to the growing medium; NAA + TZ: 0.3 µM NAA and 0.1 µM *t*Z added to the growing medium. (**b**) Rooting capacity of ‘Micro-Tom’ explants at 17 dae. (**c**) Representative images of the hypocotyl regions in the studied explants at 21 dae. (**d**) Shoot regeneration stages of ‘Micro-Tom’ explants at 17 dae. (**e**) Representative image of *DR5::GUS* expression in the hypocotyl explants at 48 h after Lov125 + Axl50 treatment. (**f**) AR primordia expressing *DR5::GUS* at 48 h. (**g**) Length of the distance to the wounding (L0) and of the AR formative region (L1) at 48 h. Different letters (**a**,**b**,**d**) and asterisk (**g**) indicate significant differences (*p*-value < 0.01) between the studied treatments or regions, respectively. Scale bars in c, e: 1 mm.

## Data Availability

Raw sequence files and read count files are publicly available in the NCBI’s BioProject repository (PRJNA731333). Gene functional annotation is available in the Appendix A of this article. All other data that support the findings of this study are available from the corresponding author upon reasonable request.

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
