# Peer review of "Dynamic Hormone Gradients Regulate Wound-Induced de novo Organ Formation in Tomato Hypocotyl Explants"

_ijms, 2021, doi:10.3390/ijms222111843_

Round 1

Reviewer 1 Report

Major comments are related to the Figures. 

Figures contain many panels. I highly suggest the reduction to a maximum of 6 panels per figure and put the others as supplementary figures:

  • Graph bars in Figures 2b and 4a are confusing. I suggest using colors to differentiate the hormonal levels in regions.
  • Figure 2e must be 2f and viceverce
  • Figures 3f & 7b could be supplementary or increase the size of letters. Maybe another type of representation could be more informative.
  • All the Log2FC values in Figures 3g, 5a & 5b are statistically significant? I suggest adding "*" to p<0.001 values (or your cut-off value).
  • Figure 5d color code was lost.
  • I suggest including a final integrative visual representation (as supplementary) that includes all the models proposed in the conclusions.

Reviewer 2 Report

The authors presented a solid experimental design conducted with the purpose of analyzing how dynamic hormone gradients interfere with wound-induced de novo organ formation in tomato hypocotyl explants. That experimental design includes phytohormone levels quantification, analyzed in different stages, and correlated with targeted profiling of phytohormone-related metabolites, and in my opinion it fits the scope of International Journal of Molecular Sciences.

My only suggestion to the authors is that figure legends should be improved. Figure legends should be self-explanatory, so that the reader could interpret them without the need of further information, and often explanation on abbreviatures and graph titles are missing on the figures presented. Due to the complexity of some of the figures, I believe this would improve the quality of the manuscript.

Besides this minor aspect, I believe the results are well presented and support the conclusions, and consider that the manuscript can be accepted after minor revisions.
